# What Drives Innovation: The Canadian Touch on Liposomal Therapeutics

**DOI:** 10.3390/pharmaceutics11030124

**Published:** 2019-03-16

**Authors:** Ada W. Y. Leung, Carolyn Amador, Lin Chuan Wang, Urmi V. Mody, Marcel B. Bally

**Affiliations:** 1Cuprous Pharmaceuticals Inc., Vancouver, BC V6T 1Z4, Canada; mbally@bccrc.ca; 2Department of Chemistry, University of British Columbia, Vancouver, BC V6T 1Z1, Canada; 3Experimental Therapeutics, BC Cancer Research Centre, Vancouver, BC V5Z 1L3, Canada; carolyn.amador13@gmail.com (C.A.); lwang@bccrc.ca (L.C.W.); uvijay@bccrc.ca (U.V.M.); 4Pathology and Laboratory Medicine, University of British Columbia, Vancouver, BC V6T 2B5, Canada; 5Pharmaceutical Sciences, University of British Columbia, Vancouver, BC V6T 1Z3, Canada

**Keywords:** liposomes, drug delivery systems, innovation, lipid nanoparticles, Metaplex

## Abstract

Liposomes are considered one of the most successful drug delivery systems (DDS) given their established utility and success in the clinic. In the past 40–50 years, Canadian scientists have made ground-breaking discoveries, many of which were successfully translated to the clinic, leading to the formation of biotech companies, the creation of research tools, such as the Lipex Extruder and the NanoAssemblr™, as well as contributing significantly to the development of pharmaceutical products, such as Abelcet^®^, MyoCet^®^, Marqibo^®^, Vyxeos^®^, and Onpattro™, which are making positive impacts on patients’ health. This review highlights the Canadian contribution to the development of these and other important liposomal technologies that have touched patients. In this review, we try to address the question of what drives innovation: Is it the individual, the teams, the funding, and/or an entrepreneurial spirit that leads to success? From this perspective, it is possible to define how innovation will translate to meaningful commercial ventures and products with impact in the future. We begin with a brief history followed by descriptions of drug delivery technologies influenced by Canadian researchers. We will discuss recent advances in liposomal technologies, including the Metaplex technology from the author’s lab. The latter exemplifies how a nanotechnology platform can be designed based on multidisciplinary groups with expertise in coordination chemistry, nanomedicines, disease, and business to create new therapeutics that can effect better outcomes in patient populations. We conclude that the team is central to the effort; arguing if the team is entrepreneurial and well positioned, the funds needed will be found, but likely not solely in Canada.

## 1. Perspective

Reviews are biased and this one is no exception. The senior author of this review completed his PhD training in the laboratory of Pieter Cullis, an innovator and serial entrepreneur. Although the PhD research was focused on gaining a better understanding of lipids in membranes, the outcome of this research changed the senior author’s research directions and highlighted the importance of the team, solutions-oriented thinking, entrepreneurialism, and determination. In the past 30 years, the senior author has been pursuing his research under a common theme: There was never enough money. Yet the team that Cullis created chose to take on challenges that many told us were misguided. The underlying message—if one does not take on the task oneself, then it is very likely to never move forward and be successful. Of course, it is necessary to define success, which, from the perspective of the senior author (an academic by nature), is when the research efforts of trainees and collaborators touch a human. In this context, success can only be made in teams that were not intimidated by the initiation of companies that recognized innovative capabilities and captured intellectual property while continuing despite rejection. Money, always being an issue, is secondary. The question now is how success can be achieved faster and more frequently, noting that delays and innovation are mutually exclusive events.

## 2. A Brief History

The concept of liposomes was first described by Alec Bangham in the 1960s when he demonstrated the spontaneous assembly of egg lecithin into multilayer vesicular structures as phospholipids were introduced to aqueous solutions [1]. Liposomes first garnered scientific interest because of their structural similarity to cellular membranes [2]. This led to extensive studies exploring membrane structure, permeation, adhesion, and fusion as well as the roles of lipids within biological membranes [1,2]. Some of these works were pioneered by the Cullis group at the University of British Columbia (UBC) in collaboration with Ben de Kruijff (Utrecht University): Investigators who are internationally recognized for their discovery of lipid polymorphisms and the various behaviours of membrane phospholipids [3,4,5]. In the early 1970’s, many compounds, such as lysozyme, chlorophyll a, and beta-fructofuranosidase, were investigated as candidates for liposomal encapsulation [6,7,8]. At this time, it became clear that certain compounds were not stored in the internal aqueous compartment; rather, they were associated with the lipid bilayer, suggesting that these compounds could become associated with liposomes by interacting with the hydrophobic regions of liposomes or by simple binding to the lipid membrane [6,7]. Additionally, it was found that encapsulation in liposomes resulted in localized cargo delivery [8]. This provided motivation for further investigation on compounds, such as actinomycin D and penicillin, by early pioneers, like Gregoriadis et al. from the United Kingdom [9]. His group also investigated the use of liposomes to carry other enzymes, as liposomes had the potential to protect enzymes from protease in the serum as well as the body’s immune response [10,11]. Later, it was observed that packaging antigens into positively-charged liposomes lowered immune responses, suggesting that liposomal formulations could be key to preventing the development of severe allergic reactions [12,13]. Eventually, researchers became interested in examining the pharmacokinetic properties of liposomal drug formulations. Arakawa et al. used encapsulated 131I-insulin, 14C-sucrose, 14C-inulin, and 14C-cefazolin sodium as markers to evaluate the liposome elimination rate [14]. It was observed that drug-containing liposomes were eliminated more slowly than the unencapsulated “free” drug. However, drug absorption was also delayed, as the release of the drug from the liposome depended on the liposome’s membrane composition and the loss of its structural integrity over time. To further complicate matters, it became apparent that the liposome composition, dose, size, and charge all affect the rate of elimination from the bloodstream [15,16]. These findings constitute the initial understanding of how liposomes interact with the body when given intravenously.

Anti-cancer drugs have been commonly selected for liposomal formulations, often in an attempt to reduce their toxic effects while maintaining or even enhancing antitumor activity. Initial attempts include the work by Steerenberg et al. demonstrating that the nephrotoxicity of cisplatin (CDDP) was decreased when the compound was encapsulated in liposomes. However, not only did antitumor activity decrease upon encapsulation, the tumors recurred and resistance to CDDP developed [17]. In contrast, the work by Sharma et al. showed a drastic increase in potency against models of ovarian cancer when *N*-(phosphonoacetyl)-l-aspartate was encapsulated in liposomes [18]. Aside from the typical preparation of liposomes for parenteral administrations [14,15,19], topical formulations were considered and these reduced the encapsulated drug’s side effects due to the liposome’s ability to increase the drug concentration at the target site while decreasing the drug exposure at off-target sites that often suffer from adverse effects [20,21]. One such example is the work completed by Harsani et al. from Michael Mezei’s group in Dalhousie University, where they demonstrated that a liposomal formulation of radioactive triamcinolone acetonide palmitate (^3^H-TRMAp) could be used as an effective topical treatment for oral ulcers [21]. Similarly, localization of the drug in the desired area improved the local anaesthetic effect of lidocaine when it was applied as a liposomal formulation instead of the conventional cream [22]. The preparation of lidocaine liposome for skin delivery continues to be of interest based on the recent development of penetration enhancer-containing vesicles by Fadda’s group in Italy [23,24]. Liposomal antibiotics and antiviral drugs have also been successfully used in intravitreal applications to treat *Propionibacterium acnes* endophthalmitis and cytomegalovirus retinitis [25]. The years between 1980–2000 were fruitful in the context of liposomal pharmaceuticals as numerous products received regulatory approval for the treatment of cancer (Doxil^®^, DaunoXome^®^, Depocyt^®^, Myocet^®^), infectious diseases (Abelcet^®^, Ambisome^®^, Amphotec^®^), macular degeneration (Visudyne^®^), as well as for the prevention of viral infections in the form of virosomal vaccines (Epaxal^®^ and Inflexal^®^) [26].

The development and advancement of liposomal technologies has proven successful in part due to the development of liposomal pharmaceuticals, but their real impact has been felt in the cosmetic industry, with products including Capture (C. Dior), Niosomes (L’Oréal), Revision (Revision SkinCare), and others. This review, however, will focus on Canada’s contribution to the liposomal pharmaceutical field as a number of liposomal technologies have been developed in Canada, ranging from methods of liposome preparation to drug loading strategies, storage strategies, and targeted delivery. This focus is on how Canadian investigators and entrepreneurs impacted the field, but the success of this technology is global. We hope those reading this paper accept its focus and understand that we needed to neglect many other key individuals that made this field what it is today: Particularly teams in the USA, Japan, the Netherlands, and Israel. A timeline of the highlighted technologies is provided in Figure 1.

## 3. Technologies for the Production of Liposomes

While liposomes are known to consist of phospholipids that self-assemble into multi-layer vesicles, uniformity of the liposomal structure is necessary for further pharmaceutical development. Liposomes for pharmaceutical applications are typically up to 200 nm in diameter, composed of a unilamellar or bilamellar bilayer and an aqueous core (Figure 2) [27,28]. Preparation of these homogeneous liposomal formulations was pioneered by Olson et al. from the laboratory of one of the pioneer liposomologists, Demetrios Papahadjopoulos (University of California San Francisco), where multilamellar vesicles were sequentially passed through polycarbonate membranes of 1.0, 0.8, 0.6, 0.4, and 0.2 µm pore sizes to yield a homogeneous preparation of liposomes with a mean diameter of about 270 nm [29]. Extrusion using this method could be completed using 10–12 mM lipid suspensions at low pressures (about 50 psi). While this method was sufficient to generate bench-scale formulations, it was challenging and time-consuming to prepare larger batches, which would be required for preclinical or clinical studies. Hope et al. from Pieter Cullis’ group at UBC further advanced the extrusion technology: Lipid concentrations up to 300 mM could be used to extrude multilamellar vesicles through 100 nm pore size polycarbonate filters [30,31]. Using higher pressure (up to approximately 500 psi), unilamellar vesicles with a mean diameter of 60–100 nm could be produced within 10 min. This invention led to the creation of the Lipex^®^ Extruder, which was first marketed by the spin-off company, Lipex Biomembranes, created in 1984. Lipex Biomembranes was bought by Northern Lipids, another biotechnology venture created in Vancouver, Canada. Northern Lipids was eventually purchased by Evonik Industries, a German multinational company. Evonik still retains the Northern Lipids enterprise in British Columbia and still markets the Lipex^®^ Extruder today. This extrusion technology, ranging from simple devices for laboratory scale production to larger extrusion systems that can handle more than 100 L, continues to be the industrial standard for preparing pre-clinical and clinical batches of liposomal formulations.

As the technology of microfluidics emerged in the last two decades, novel methods of manufacturing liposomes have also been developed. Microfluidics systems manipulate and control the flow of fluids through networks of channels having cross-sectional dimensions of approximately 5 to 500 µm [32]. The first use of microfluidics mixing to generate liposomes was described by Jahn et al., where they demonstrated that hydrodynamically focusing of alcohol-dissolved lipids between two sheathed streams of aqueous buffer in a microfluidic channel could yield monodispersed liposomes ranging from 50 to 150 nm depending on the flow rate [33]. To better control the mixing process and to generate liposomes even more rapidly, Zhigaltsev et al. from the Cullis group here in Canada designed a microfluidic mixing device based on the concept of staggered herringbone mixing [34,35,36]. Using this method, liposomes of 20–50 nm in size could be prepared and loaded with small molecules, such as doxorubicin [35]. This work led to the creation of the NanoAssemblr™ device, which is commercialized by Precision NanoSystems (Vancouver, BC, Canada) for the preparation of liposomes encapsulating small molecules as well as macromolecules, such as nucleic acids and proteins, at bench and production scales [34,37]. This device is particularly well-suited for the self-assembly process used to prepare cationic lipid/anionic polymer (e.g., DNA, RNA, antisense oligonucleotide, siRNA, peptides, etc.) complexes, often referred to as lipid-nanoparticles (LNP). The ability of anionic polymers to bind cationic lipids to form complexes was first disclosed by Dr. Bally’s team [38,39,40], and was later employed by Cullis and associates to define what is now a US Food and Drug Administration (FDA) approved siRNA formulation called Patisiran. It has been postulated that these structures are better defined as a particle, rather than a liposome because they likely do not comprise a lipid-bilayer structure.

## 4. Technologies for the Storage of Liposomes

Another practicality issue associated with the pharmaceutical development of liposomal formulations was the shelf-life of the products. Liposomal formulations typically require storage at 4 °C and are relatively unstable for long term (>2 years) storage compared to other pharmaceuticals that can be prepared as dried products. A discovery by J. Crowe, Louis Crowe, and Dennis Chapman overcame this problem. Crowe’s team were trying to better understand membrane stability in the presence of carbohydrates, known to be secreted by anhydrobiotic organisms (e.g., nematode) that were able to survive drying/freezing conditions. They showed, in 1984, that carbohydrates, such as trehalose and sucrose, were able to stabilize the model membrane structure at low water contents [41]. This observation was applied by Madden et al. working in Vancouver, and it was shown that multiple types of sugars, including trehalose, sucrose, and lactose, could be effective at protecting liposomes during the dehydration-rehydration process when the sugar content was appropriately adjusted [42,43,44,45]. The first publication on this was released in July of 1985 by the Vancouver team, but was followed shortly thereafter by a publication from Crowe et al. in October of that year, showing that trehalose can be used to prevent liposomes from fusing during the freeze-drying process [41]. The use of carbohydrates to protect liposomes was disclosed in a patent with the Vancouver inventors and the technology was commercialized and functionalized with lactose being incorporated as a cryoprotectant in Amphotec^®^, Myocet^®^, and Visudyne^®^, and sucrose being added to AmBisome^®^ to enable the preparation of lyophilized liposomal products. Perhaps most interesting, this approach worked very well in liposomes with no or low (<20 mol%) levels of cholesterol, where the ability to prevent aggregation and fusion during a freeze/thaw cycle was first demonstrated, even in the absence of carbohydrates, by Dos Santos et al. [46,47]. The first low-cholesterol product, Vyxeos^®^, is stored as a dehydrated powder using sucrose and has a shelf life of 2 years at 4 °C to 8 °C.

## 5. Optimization of Liposomes for Pharmaceutical Use

A summary of the main approaches discussed in this section is listed in Table 1.

### 5.1. Improvement of Encapsulation Efficiency of Passive Loading

Following the initial forays into liposomal drug delivery, it became increasingly clear that more efficient ways to encapsulate drugs were needed. Typical drug entrapment efficiencies were at best 10% and this was due to a number of limitations [65,66]. In particular, given an optimal size of 100–200 nm for nanomedicines that attempt to leverage the enhanced permeability and retention (EPR) effect, a trapped volume of 1.5–2.5 µL/µmole, and a workable lipid concentration of 10–20 mM, it was just not possible to achieve trapping efficiencies above 10% [67]. Of course, methods that used increased lipid concentrations and/or association of the drug candidate with the membrane could be designed to achieve an improved trapping efficiency. In some cases, increasing the aqueous drug solubility of the compound of interest by changing the pH of the medium in which the compound is suspended or by increasing the temperature greatly enhanced the encapsulation efficiency. Regarding lipid concentration, the high pressure extrusion method allowed for the manufacturing certain liposomes at lipid concentrations of 300 mM and this could achieve trapping efficiencies as high as 80% [30]. Alternatively, in 1984, Kirby and Gregoriadis introduced the dehydration-rehydration method, where dehydrated liposomes were rehydrated in a small volume to increase the drug encapsulation efficiency to as high as 40–50% [48]. As suggested above, depending on the hydrophobicity of the compound being used, its association with the lipid membrane could be increased or the membrane’s “fluidity” could be altered to enhance the association of these drugs [46,49,66]. However, these strategies came at a cost: Low drug-lipid ratio, which in one aspect meant a great deal of liposomal lipid was required to administer an effective dose of the therapeutic agent. Given the cost of lipids and drugs at the time, this made it unreasonable to pursue liposomal drugs as pharmaceuticals.

### 5.2. Development of Remote Loading Methods

In 1976, Nichols and Deamer demonstrated that catecholamines can accumulate within liposomes that have an established transmembrane pH gradient [68]. This concept was confirmed in 1985 by Bally et al., who discovered that lipophilic cations, like safranine O, could accumulate inside liposomes in response to an Na^+^/K^+^ electrochemical gradient where the liposome’s interior was negative [50]. This resulted in an interior safranine concentration over 80 mM, many times greater than the solubility of the safranine. Research completed by Deamer’s (USA) and Cullis’ (Canada) groups set the foundation for remote or active loading, where it was possible to achieve a >98% encapsulation efficiency [46,50,51,52,53]. The pH gradient loading method, and varieties thereof, remains to be one of the most employed methods to encapsulate a drug or drug candidate in liposomes. Several methods for creating these gradients exist, such as using citrate buffer in the aqueous compartment, using an ionophore-mediated ion gradient to generate a pH gradient as originally described by the Vancouver team [54,69,70], or using transmembrane ammonia gradients as described by the Israel/US teams. Depending on the properties of the compound of interest, the Canadian team led by Marcel Bally showed that encapsulation efficiencies could be further improved by the addition of solvents, such as ethanol. While the use of ethanol could potentially double the encapsulation efficiency at lower temperatures, an excessive amount of ethanol could cause the liposomal structure to break down and collapse the pH gradient [46].

Although remote loading of small molecules in response to a transmembrane pH gradient has been widely applied, three issues still remain to be addressed: (1) Many small molecules do not have a protonizable amine function, which is required for efficient pH gradient loading; (2) some compounds, which are amenable to pH gradient loading, are associated with poor trapping efficiencies, which could be due to issues, such as proton leakage through the bilayer [53,71]; and (3) many therapeutically interesting compounds are poorly soluble in aqueous solution, but are not necessarily “hydrophobic”. To increase drug encapsulation efficiency for various types of compounds, new encapsulation methods have been established. One was referred to as a microencapsulation method, in which a water/organic/water emulsion is agitated and used to prepare a liposomal suspension [55]. Other approaches to increasing the stability of liposomes include the development of “layersomes”, where multiple layers of polyelectrolytes are added to conventional liposomes; this proved effective for the encapsulation of piroxicam [56]. Liposomes, functionalized with hyaluronanic acid, have also been produced and were found to increase the bilayer packing order, reducing membrane flexibility and improving drug penetration in topical applications [57].

Another strategy that has been employed to improve the trapping efficiency of small molecules is the use of metal complexation. Initial studies were completed by the Cullis group in which doxorubicin was encapsulated into Sphingomyelin/Chol liposomes in response to a manganese (trapped MnSO_4_) ion gradient [58]. Greater than 98% trapping efficiency was achieved, but the stability of the MnSO_4_ solution required the use of a low pH. Although it was suggested that doxorubicin was capable of forming a metal complex with Mn^2+^, the fact that the liposomes used also had a pH gradient confused the interpretation of the results. The observation was confirmed by Abraham et al. from the Bally group [59] in a manner that allowed that group to conclude that metal complexation could be the sole driver of encapsulation. Further, the technology appeared suitable for use with a number of other different metal ions, including copper and zinc [72], but the method always relied on the use of compounds that exhibited a solubility >1 mg/mL and the use of compounds that had a protonizable amine. The role of the pH gradient versus metal to encapsulate the drug was further confused by the Bally team, who discovered and patented that transition metals could be used in conjunction with a divalent metal ionophore (A23187) to generate a pH gradient. However, there was something about the use of copper that enhanced drug retention that was surprising and unexpected [73]. The product generated using this technology was focused on the camptothecins (irinotecan and topotecan) [54,74,75] and one of the resulting products, Irinophore C, was more active than any other previously described irionotecan formulations. It was disappointing that the resultant formulation never made it to the clinic. There are many reasons for this, including the development of another formulation of irinotecan (now approved and called Onivyde^®^ [76]) that was clinically more advanced than the one created in Canada and the fact that the technology developed in Canada was licensed: The company that licensed the technology was not in a position to develop the product. The resulting delays and the early filing of intellectual property (which was granted in several countries) made it unlikely that further investments in funding clinical trials would result in meaningful returns to those that made that investment. The Bally’s group was too slow to develop the technology.

It is worth noting that the incorporation of a transition metal into liposomes was critical to the creation of Vyxeos^®^: A copper-containing formulation wherein the formation of a copper-anthracycline complex is used to reduce the leakage rate of daunorubicin, while the use of low cholesterol liposomes was required to enhance the retention of cytrarabine. The resulting product was designed such that the two cytotoxic agents could be released from the liposomes at identical rates, ensuring the maintenance of a synergistic drug-to-drug ratio in vivo [77].

This transition metal-complexation technology has further evolved through works completed by the Bally’s group, who is now working with other founders (Ada Leung and Thomas Redelmeier), the Vice President of research (Michael Abrams), and the entrepreneurial group at UBC (e@UBC and HATCH) to lead the development of what is referred to as Metaplex technology. The Metaplex technology is an active loading platform wherein a transition metal ion gradient is established across the membrane and used as the primary driving force to accumulate drugs inside liposomes; drugs that exhibit limited water solubility, may not contain a protonizable amine, but do contain a metal binding function [60,61]. In this technology, there is clear evidence that the selected drug has a metal binding function. By using lipid nanotechnology and metal coordination chemistry, this new formulation method created by the Bally group enables the development of drugs that are typically relegated to medicinal chemistry groups. Initial studies by Wehbe et al. explored formulation strategies for diethyldithiocarbamate (DDC), a metabolite generated after disulfiram is administered. Disulfiram is an anti-alcoholic agent known to inhibit aldehyde dehydrogenase [78]. DDC has long been known as a copper binding compound [79] and it has been shown to have copper-dependent anticancer activity [80,81,82]. Metaplex technology has been further expanded at the Vancouver-based Cuprous Pharmaceuticals Inc. for two different classes of compounds: (1) Sparingly soluble small molecules that have metal-coordinating motifs, which could benefit from drug delivery technology; and (2) relatively inactive small molecules that become therapeutically active upon complexation with metal ions, such as Cu^2+^. The preparation of nanoformulations for metal-dependent therapeutics was originally inspired by the increase in the number of publications in recent years demonstrating the therapeutic activity of copper complexes against a variety of disease indications, including cancer, inflammatory diseases, neurodegenerative diseases, and infectious diseases [83,84,85,86,87,88]. While metal complexes, or specifically copper complexes, hold promise as therapeutic agents based on in vitro data, there is a lack of preclinical evidence supporting their utility. The major reason for this is likely due to the fact that most of these therapeutically active metal complexes have poor solubility in aqueous solutions under physiological conditions, making it a challenge to test these agents in animals. The Metaplex technology addresses this problem by using liposomes as nano-scale reaction vessels in which metal coordination occurs [60]. Furthermore, these nanoparticles can be suspended in biocompatible aqueous buffers for parenteral administration into relevant preclinical models. The Bally group and Cuprous validated the concept of metal-dependent anticancer activity for copper-coordinating compounds through an in vitro screen on platinum-sensitive and platinum-resistant cell lines and prepared injectable copper-based formulations of DDC and clioquinol [80,89,90]. The Metaplex technology can also be used to reformulate sparingly soluble compounds that have metal-binding properties with the goal of either reducing toxicity or improving therapeutic activity. This was demonstrated by preparing liposomal formulations, CX-5461, an investigational compound that interacts with copper and when formulated using Metaplex technology, is more efficacious than the low pH clinical formulation currently used [91,92]. Cuprous is currently developing this technology for immuno-oncology treatments reliant on the use of small molecules rather than the more expensive antibody-based or cell-based therapeutics. Immunogenic cell death (ICD), a phenomenon wherein dying cancer cells emit specific molecular signals that ultimately lead to an anti-tumour adaptive immune response followed by long-term protection against recurrence, is a concept of immunotherapy that has recently garnered much attention due to its potential to treat metastatic disease and/or bring about long-term survival or cures [93,94]. Anthracyclines, like doxorubicin, are known to induce ICD as a secondary mechanism [93,95]. Most interestingly, metallic copper itself, is known to generate reactive oxygen species (ROS) and induce endoplasmic reticulum (ER) stress, which is required for ICD induction [96,97]. Cuprous is exploring exciting new opportunities for enhancing the delivery of such compounds using its proprietary platform technology [73,74].

Metaplex has the potential to work with a more diverse array of compounds than conventional pH gradient loading due to a larger chemical space that would satisfy the requirement for metal-ligand coordination to occur [86,87]. While the focus has thus far been on the use of Metaplex for oncology-based formulations, the potential application of this technology is much broader. Copper complexes of non-steroidal anti-inflammatory drugs (NSAIDs) have been shown to be associated with reduced toxicity and increased therapeutic activity [88,98]. Various copper complexes have exhibited hypoglycemic effects and may be suitable as diabetic treatments [99,100]. Other studies demonstrated that copper complexes could have potent antimicrobial activity and could be useful against infections by superbugs, which are becoming a global health concern [101,102]. Finally, it is known that metal imbalance is strongly associated with neurodegenerative diseases, such as amyotrophic lateral sclerosis, Alzheimer’s, and Parkinson’s diseases [83,103,104]. Strategies to adjust these imbalances using metal chelators are being evaluated [105,106]. All of these represent opportunities for the development of novel metal-based therapeutics, where the Metaplex platform could be used to design formulations for specific indications, further expanding the application of liposomal technologies non-oncology-based indications.

### 5.3. Development of Liposomes for Encapsulation of Nucleic Acids

The utility of liposomes has also been extended to nucleic acid delivery (DNA, mRNA, Antisense oligonucleotides, siRNA, etc.), typically for the purpose of genetic modification of target cells. Nucleic acids alone cannot pass through cellular bilayers, but liposomes have been designed to fuse with membranes and successfully deliver associated payloads [107]. The earliest examples of nucleic acid encapsulation were in the late 1970s: Dimitraidis et al. encapsulated mRNA, rRNA, and tRNA in large unilamellar liposomes, and Fraley et al. focused on the delivery of pBR322 bacterial plasmids [108,109]. Continued research demonstrated that factors, such as the presence of polyethylene glycol (PEG) or glycerol, liposome charge, and the number of lamellae, all affect nucleic acid infectivity or sequestration [110]. While early work supported the use of cationic liposomes as delivery agents of plasmid DNAs for transfection purposes, the physical characteristics of these liposome/DNA complexes were not well-defined. In 1995, Reimer et al. from the Bally group prepared and characterized, for the first time, hydrophobic complexes of cationic lipids and plasmid DNA that can be readily extracted in organic solvents [39,40]. They proposed these cationic lipid/DNA complexes as potential intermediates for the formation of particles suitable for gene delivery to cells [38,39,40]. This work was extended to the complexation of cationic lipids with antisense oligonucleotides designed for gene silencing and subsequently the addition of PEG to prevent aggregation of these lipid/nucleic acid complexes [111,112,113]. These works were further developed in the 1990s and 2000s through the Cullis group (UBC) and Vancouver-based companies, including Acuitas Therapeutics (formerly AlCana Technologies) and Inex Pharmaceuticals (now Arbutus Biopharma) [64,114]. These efforts led to the use of ionizable amino lipids for the delivery of nucleic acids [63] and the development of fusogenic liposomes: Liposomes that have an exchangeable PEG-lipid conjugate, which contributes to the in vivo stability of nanoparticles, particularly those encapsulated with antisense oligodeoxynucleotides [115,116]. These technologies seeded the evolution of nucleic acid drug delivery, leading to the creation of new lipids designed specifically for the encapsulation of RNA-based therapeutics and the development of the lipid nanoparticle (LNP) technology platform: The most advanced and currently the only clinically validated nucleic acid delivery system through the regulatory approval of the RNA interference (RNAi) therapeutic Onpattro^®^ [5,62,64,114,117,118,119,120].

## 6. Other Key Canadian Discoveries that Impacted the Development of Therapeutically Interesting Drugs

Table 2 below highlights some of the Canadian discoveries that helped the development of liposomal pharmaceutical products evolve over time.

### 6.1. Selective Drug Delivery with Liposomes

Shortly after liposomes were first described, their promise as selective delivery agents was considered. To this end, Gregoriadis et al. associated molecular probes to drug-containing liposomes and found that probes (Immunoglobulin G’s (IgGs) raised against different types of cells) could mediate selective cellular uptake [125]. In the early 1980s, Leserman et al. coupled monoclonal antibodies to liposome surfaces, successfully demonstrating cell-specific liposome interaction [126]. A third example by Guru et al. demonstrated significant increases in the efficacy of sodium stibogluconate via encapsulation in tuftsin-bearing liposomes. Even liposomes carrying only tuftsin were found to make animals resistant to *Leishmania donovani* infections [127]. In addition to these early studies, Ryman’s team explored the imaging potential of liposomes, highlighting the ability of liposomes to localize in lymph nodes by injecting technetium-99m labelled liposomes into rats and then studying tissue distribution via γ-camera imaging and radioassay [128], and Morgan et al. demonstrated that liposomes could be used to image staphylococcal infections [129]. Finally, Baldeschwieler’s group highlighted the potential for liposomes to image tumours. All these studies were completed well before Matsumura and Maeda first published and described the EPR effect: Selective accumulation due to abnormally permeable vasculature found in tumours [130]. It is also worth noting that drug release mechanisms, such as endocytosis and fusion, were investigated early in the development of liposomal pharmaceuticals, where it was postulated that different uptake mechanisms could allow for selective delivery depending on how specific liposome formulations interacted with cells [65].

Clearly, an important rationale for developing more selective liposomes was based on strategies designed to increase interactions between the nanoparticles and target/disease cells while minimizing toxicities against healthy cells. One of the most commonly employed approaches still used today concerns the use of surface coatings. Liposomes with attached antibodies could bind to specific cell populations. Some of the earliest works by Papahadjopoulos’ group demonstrated that liposomes coated with antibody fragments or immunoliposomes were able to bind human erythrocytes much more efficiently compared to non-targeting liposomes [131,132]. In Canada, some of the early studies investigating the use of antibody-mediated targeting of liposomes to treat cancer were conducted by Theresa (Terry) Allen’s group at the University of Alberta [121]. For instance, Ahmad and Allen demonstrated that liposomes coated with antibodies targeting squamous carcinoma cells resulted in increased uptake and cytotoxic effects against KLN-205 lung squamous carcinoma cells relative to non-targeting liposomes [133]. Several studies have demonstrated that actively targeted liposomes may contribute to improved therapeutic activity in vivo [123,134,135]. While the initial focus was to alter the biodistribution of targeted liposomes for more efficient delivery to the target site (i.e., the tumour), it was discovered that delivery to the tumour for both immunoliposomes and conventional liposomes was primarily dependent on the EPR effect [136]. The differences in therapeutic activity reported were likely due to an increased uptake of immunoliposomes by cancer cells as a result of receptor-mediated endocytosis followed by the escape of the cytotoxic agent from endosomal/lysosomal degradation [136,137,138,139]. In recent years, researchers have also explored the use of peptide-mediated targeting [138,140,141]. Several excellent review articles are available describing the various functionalization strategies that have been employed in the development of active targeting nanomedicines [142,143,144]. Although first envisioned 40 years ago, there has yet to be a successful targeted formulation approved for clinical use. However, it is notable that the limitations for targeting solid tumours are clear. Allen’s team was able to highlight the potential of targeting liposomes to cells within the vascular compartment [123]. It is hoped that this may prove to be of therapeutic value, particularly in light of some of the recent findings from the Bally group, which emphasize that therapeutic antibodies may exhibit improved therapeutic effects when attached to liposomes. These studies consider the potential of liposomes to deliver antibodies rather than antibodies to target liposomes [123].

### 6.2. The “PEGylation” Technology

The effects of various lipid compositions on the pharmacokinetics of liposomes were also explored with the goal of prolonging the presence of liposomes in the plasma compartment. The first attempt was made by Terry Allen’s group through the addition of GM_1_ ganglioside into liposomes, which reduced mononuclear phagocyte system (MPS) uptake, allowing the liposomes to remain in the blood stream for several hours [145,146,147]. Based on this pioneering work, scientists explored the incorporation of PEG into formulations as a steric stabilizer lipid (i.e., 1,2-Distearoyl-sn-glycero-3-phosphoethanolamine-polyethylene glycol (DSPE-PEG)) [147]. This “PEGylation” technology, otherwise known as stealth liposome technology, has been the most widely employed strategy since the 1990s [26,148,149,150]. Yet the role of surface coating of liposomes with PEG has likely been well overstated. It was the Canadian Bally group along with Christine Allen, Nancy Dos Santos, and others that first highlighted that the primary role of PEG on the surface of the liposomes was not to prevent protein association or even association with phagocytic cells, but to prevent surface–surface interactions that could lead to aggregation of liposomes [124]. This was elegantly proven by the work of Dos Santos, who demonstrated that selected liposomes could be prepared in the absence of cholesterol or low levels of cholesterol as long as they incorporated lipids that prevented their aggregation, such as PEG-modified lipids [124]. As suggested above, this technology was key to the development of Vyxeos^®^, a combination liposomal drug product now approved for treatment of acute myeloid leukemia.

### 6.3. Strategies to Encapsulate Multiple Agents

Related to the previous statement and existing evidence that cancer is a heterogeneous disease, which is most effectively treated with a combination of multiple therapeutic agents, there was significant interest in encapsulating multiple drugs in the same liposome. For example, daunorubicin and 6-mercaptopurine are a pair of chemotherapeutic compounds that were thought to act synergistically—one being a hydrogen acceptor and the other being a hydrogen donor. While this particular interaction was not observed, combining the two compounds in a dual drug liposome did appear to show synergistic cytotoxic effects [151]. Researchers in Vancouver (BC Cancer and Celator Pharmaceuticals) extensively studied the impact of drug-to-drug molar ratios on therapeutic outcomes in vitro. Drug combinations could result in synergistic or antagonistic treatment effects depending on the ratio used. It was logical to assume that if the effects of the anticancer drug in vitro were dependent on the drug-drug molar ratio in vitro then the same would hold true in vivo [152,153]. When encapsulating multiple drugs in the same liposome, or even different liposomes, the relative release rates of the two compounds must be considered, as the goal is to achieve an ideal ratio at which they could be administered (a fixed ratio product) and to maintain that ratio over time after administration to achieve optimal therapeutic effects [154]. Tardi et al. illustrated an example of a system in which cholesterol was used to control drug leakage rates and with this system, they were able to maintain a 1:1 synergistic ratio of irinotecan and floxuridine in vivo [72]. This observation proved to be a “patenting moment” that led to the development of liposomal combination products protected under the “Combiplex” patent [155]. This patent described the use of various drug delivery systems to be used to prepare products in a manner where the combination ratio could be maintained in vivo; a patent that first contemplated the use of daunorubicin and cytarabine as a liposomal combination product that eventually became Vyxeos [77].

## 7. The Canadian Impact on Regulatory Approved and Investigational Liposomal Formulations

Here, we provide a list of approved liposomal products and reiterate some of the information above to highlight how Canadian scientists influenced these products. An up-to-date list of all regulatory approved liposomal formulations is provided in Table 3.

### 7.1. Liposomal Formulations of Amphotericin B: Abelcet^®^ and iCo-019

Amphotericin B is an antifungal agent used to treat serious fungal infections and leishmaniasis. Multiple lipid-based formulations of amphotericin B (i.e., AmBisome^®^, Abelcet^®^, Amphotec^®^) are approved for use in various countries, of which Abelcet^®^ presents a unique formulation wherein amphotericin B is complexed with dimyristoylphosphatidylcholine (DMPC) and dimyristoylphosphatidylglycerol (DMPG) at a 7:3 molar ratio, forming ribbon-like structures (hence known as amphotericin B lipid complexes), which are believed to have contributed to its favourable toxicity and therapeutic profiles [186]. The formation of Abelcet^®^ was designed by Drs. Thomas Madden, Andrew Janoff, and Pieter Cullis at UBC. Abelcet^®^ was the first drug from the Cullis group to reach the market. It was developed by The Liposome Company in association with the Canadian Liposome Company, a wholly owned subsidiary, and was approved in 1995 for the treatment of invasive fungal infections to which patients are non-responsive or cannot tolerate conventional amphotericin B treatments. Abelcet^®^ is currently a Leadiant Biosciences product.

iCo-019 is an oral liposomal formulation developed by Kishor Wasan’s group (from UBC and University of Saskatchewan). The formulation comprises Peceol and distearoylphosphatidylethanolamine (DSPE-PEG) and the resulting oral formulation was found to reduce the number of fungal colony formation units by more than 80% relative to untreated controls [187]. While the existing intravenous formulation of amphotericin B is effective at treating invasive fungal infections, safety issues associated with parenteral administrations, such as infection at the catheter, haemolysis, and renal toxicities, are concerning [187]. The oral formulation was designed to overcome these issues as well as to address the problem of these drugs being costly and difficult to administer in remote locations, where fungal infections are more problematic. The oral amphotericin B formulation is currently being developed by the Vancouver-based company, iCo Therapeutics, which has recently announced positive Phase I data on iCo-019.

### 7.2. Liposomal Formulations of Doxorubicin: Myocet^®^ and Doxil^®^

Doxil^®^ (Caelyx^®^) and Myocet^®^ are perhaps the most well-known liposomal anticancer agents. Doxil^®^ was first approved in 1995 for the treatment of acquired immunodeficiency syndrome (AIDS) related Kaposi’s sarcoma [139]. It is now also being used to treat relapsed ovarian cancer, multiple myeloma, and locally advanced or metastatic breast cancer. Myocet^®^ is another liposomal formulation of doxorubicin, which was approved in 2000 to be used in combination with cyclophosphamide to treat metastatic breast cancer in Europe. The main difference between Doxil^®^ and Myocet^®^ lies in their lipid composition, which ultimately affects their safety, drug release, and biodistribution profiles [188,189,190]. Doxil^®^ is composed of hydrogenated soya phosphatidylcholine, cholesterol (Chol), and PEG-modified phosphatidylethanolamine (55:40:5 molar ratio) while Myocet^®^ is a non-PEGylated liposomal formulation consisting of egg phosphatidylcholine (EPC) and Chol (55:45 molar ratio). Myocet^®^ increases the circulation lifetime of doxorubicin by approximately three times relative to the free agent in mice [189,191]: An effect thought to be due to the toxicity of doxorubicin being delivered to phagocytic cells [192]. This is also known to occur for Doxil^®^ because it is prepared with lipids that enhance drug retention, resulting in an increased blood residence time of doxorubicin and increased drug delivery to the skin, which made it an ideal formulation for the treatment of skin localized cancers, like Kaposi’s sarcoma [189,193,194]. However, the increased skin delivery caused dose-limiting toxicities attributed to hand-and-foot syndrome [189]. Both liposomal formulations reduce cardiotoxicity, a major concern associated with free doxorubicin. Myocet was developed by the Vancouver group in association with the Canadian Liposome/The Liposome Company [173]. Doxil^®^ originated from research completed by groups in California and Israel, but this product was greatly influenced by Terry Allen. The product was initially developed by Liposome Technology Inc. and is now owned by Johnson & Johnson. Myocet^®^, on the other hand, is now a product owned by Teva Pharmaceutical Industries [150,195].

### 7.3. Visudyne^®^

Aside from Abelcet^®^, another liposomal formulation that was not designed for oncology use is Visudyne^®^ or liposomal verteporfin, which is a benzoporphyrin derivative that serves as a photosensitzer for photodynamic therapy in the treatment of age-related macular degeneration. The formulation consists of a mixture of DMPC and egg phosphatidyl glycerol (EPG) [196] and was designed by the Canadian scientist, Thomas Madden. Visudyne^®^ is known for its selectivity against choroidal neovasculature arising from macular degeneration while minimizing the risk of severe visual acuity loss [197]. The product was developed by QLT Inc. (a spin-off company from UBC established in 1981) and is now a product owned by Bausch & Lomb Incorporated.

### 7.4. Marqibo^®^

Marqibo^®^ is the liposomal formulation of vincristine developed to address dosing and pharmacokinetic limitations of the free agent. This formulation was designed by Bally, Mayer, and Cullis in the late 1980s and early 1990s. Although the product that arose from this research was originally owned by The Liposome Company through their agreements with UBC and the Canadian Liposome Company, the technology was eventually licensed back to a Vancouver-based start-up called Inex Pharmaceuticals. While the original product was prepared using DSPC/Chol liposomes, this product had an unacceptable storage life. This problem was overcome by using a new lipid composition of sphingomyelin and Chol in a 55:45 molar ratio [182,198]. This particular formulation exhibited a better storage shelf-life, and was associated with a surprising increase in drug retention and a profound improvement in therapeutic activity, exhibiting cures in multiple murine models of leukemia [181,199]. Marqibo^®^ was originally developed by the Canadian company, Inex Pharmaceuticals Corporation, and is now a product of Spectrum Pharmaceuticals approved (2012) for the treatment of Philadelphia chromosome-negative acute lymphoblastic leukemia in adults.

### 7.5. Vyxeos^®^

Vyxeos^®^ (formerly known as CPX-351) is the first dual-drug liposomal formulation to receive regulatory approval. It comprises cytarabine and daunorubicin packaged at a fixed 5:1 molar ratio inside 1,2-Distearoyl-sn-glycero-3-phosphocholine (DSPC)/1,2-distearoyl-sn-glycero-3-phospho-(1′-rac-glycerol) (DSPG)/Chol (70:20:10 molar ratio) liposomes [200]. Vyxeos^®^ was developed based on the original concept of the CombiPlex^®^ platform technology invented by Lawrence Mayer and Marcel Bally’s group (Vancouver), where drug combinations exhibit synergistic anticancer activity when given at certain molar ratios and drug carriers could be used to maintain those ratios in vivo [152,201,202]. This technology demonstrated that (1) two drugs can be co-encapsulated into liposomes at a fixed molar ratio and (2) liposomes can be designed to optimize release kinetics such that the optimal therapeutic ratios can be achieved and maintained in vivo [203]. In 1999, Celator Pharmaceuticals Inc., was formed (Vancouver, BC, Canada) to develop the CPX product line. CPX-351(Vyxeos^®^) received regulatory approval for the treatment of treatment-related or secondary acute myeloid leukemia (AML) and AML with myelodysplasia-related changes in 2017. Just prior to this the company was acquired by Jazz Pharmaceuticals.

### 7.6. Onpattro^®^

The most recently approved liposomal formulation is Onpattro^®^, a nanomedicine that is revolutionary in multiple ways. Onpattro^®^, also known as patisiran, consists of an siRNA targeting the production of the transthyretin (TTR) protein, packaged inside LNPs, as described above. The lipid component contains Chol, DLin-MC3-DMA, DSPC, and PEG_2000_-C-DMG at weight ratios of 6.2:13:3.3:1.6 per 2 mg of siRNA. By suppressing the production of wild-type and mutant TTR, patisiran reduces the accumulation of amyloid deposits in peripheral nerves, which would otherwise cause peripheral neuropathy [204]. Onpattro^®^ is the first and only medication approved for the treatment of polyneuropathy caused by hereditary transthyretin-mediated amyloidosis. It is also the first and currently the only RNA interference therapeutic approved. Onpattro^®^ is a product developed by Alnylam Pharmaceuticals using their proprietary siRNA and the LNP technology that originated from work completed by Jayaraman et al., including experienced Canadian scientists in the field: Pieter Cullis, Thomas Madden, Muthiah Manoharan, Steven Ansell, Jianxin Chen, and Michael Hope [64]. The development of Onpattro^®^ resulted from collaborative efforts between Alnylam Pharmaceuticals and Acuitas Therapeutics (then AlCana Technologies). All commercialization work was conducted by Alnylam Pharmaceuticals.

## 8. Conclusions

Since the first description of liposomes in 1965, our knowledge about lipids and the role of lipids in membranes has expanded enormously. With this increase in understanding came innovations and discoveries that were impactful on patient treatment outcomes and quality of life, from reductions in adverse effects to controlled-release combinatory formulations. Nearly half of all regulatory approved liposomal formulations are Canadian inventions, highlighting the efforts of liposomologists from coast to coast. Researchers from Canada and around the world will endeavour to use liposomes to increase the therapeutic activity of promising compounds, making many more efficacious nanomedicines available to patients in the years to come.

There is a common theme to the success of the liposome technology developed by Canadians. First is the recognition of innovation and an aggressive patenting strategy that can protect the idea and its use. Next comes the “do it yourself” attitude: One that is most readily expressed in the context of new company formation. Finally, come partnerships, ones that include scientists, business development personnel, quality control staff, clinical trial specialists, clinicians, etc. However, the funding to develop this technology is great and, therefore, the efforts of venture capitalists and existing companies resourced to develop and commercialize therapeutic agents of value are also needed. Whether it is necessary to have a large company developed in Canada remains a question. In this context, success could be defined by partnerships with existing companies, even those in other countries. These partnerships should be highlighted as a Canadian success. Perhaps the only negative to all this is the fact that Canadian patients may be second, third or even fourth in line to have the opportunity to participate in clinical trials and access to the drug, if approved. It is more likely that successful products will be first marketed in larger economic markets, like the USA and Europe, before they reach Canadians. Further, in the absence of really compelling data, it is sometimes difficult to adopt new technology. Myocet^®^, for example, is approved in Europe and Canada, but it is rarely used in Canada, in part because it is not marketed there. This is despite evidence demonstrated in a large patient population that Myocet^®^ does reduce the cardiotoxicity of doxorubicin and is a safer product. While Canadian access to new technology created in Canada may be limiting, the training opportunities here in Canada are fantastic, resulting in an international reputation of excellence and skills.

Finally, if one looks to the future, the strength of the liposome community as well as the drug delivery communities in general is very strong in Canada. The number of drug delivery/polymer/lipid technology based companies in operation are significant and these include the BC–based companies, Evonik, Precision Nanosystems, iCo Therapeutics, Acuitus, Sitka Biopharma, Cuprous Pharmaceuticals, Integrated Nanotherapeutics, Genevant Sciences, as well as Nanostics Precison Health (Alberta) and Nanovista (Ontario). The technology is creating jobs, training highly qualified personnel, and, most importantly, creating new products that are improving the health care of patients internationally.

## Figures and Tables

**Figure 1 pharmaceutics-11-00124-f001:**
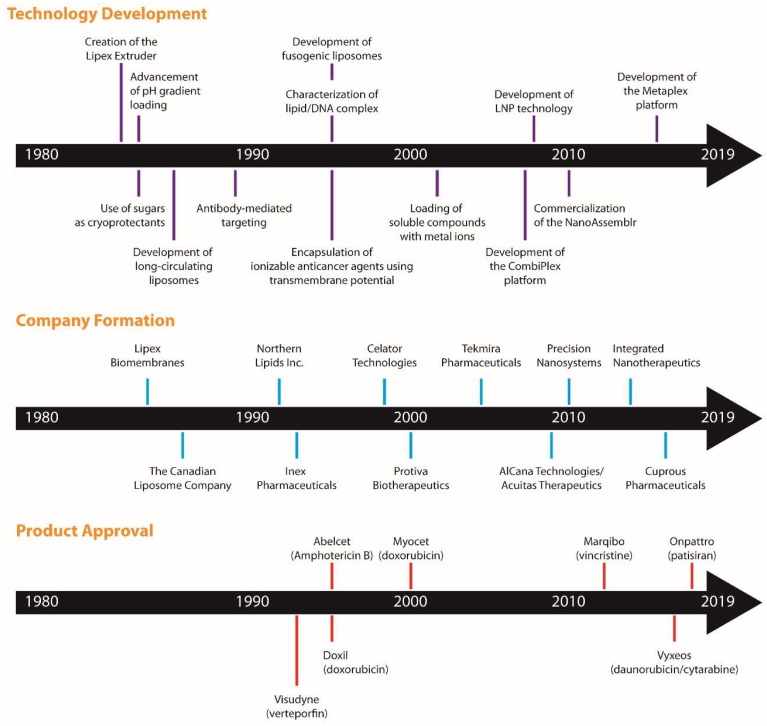
Canadian contribution to the development of liposomal technologies, formation of companies, and development of clinically approved formulations. Selected liposomal technologies are listed on the timeline based on the patent literature (top panel). These technologies led to the formation of companies, which are shown based on the year when they were established (middle panel). Regulatory approved liposomal formulations that were developed by Canadian researchers are shown on the timeline based on their year of approval (bottom panel).

**Figure 2 pharmaceutics-11-00124-f002:**
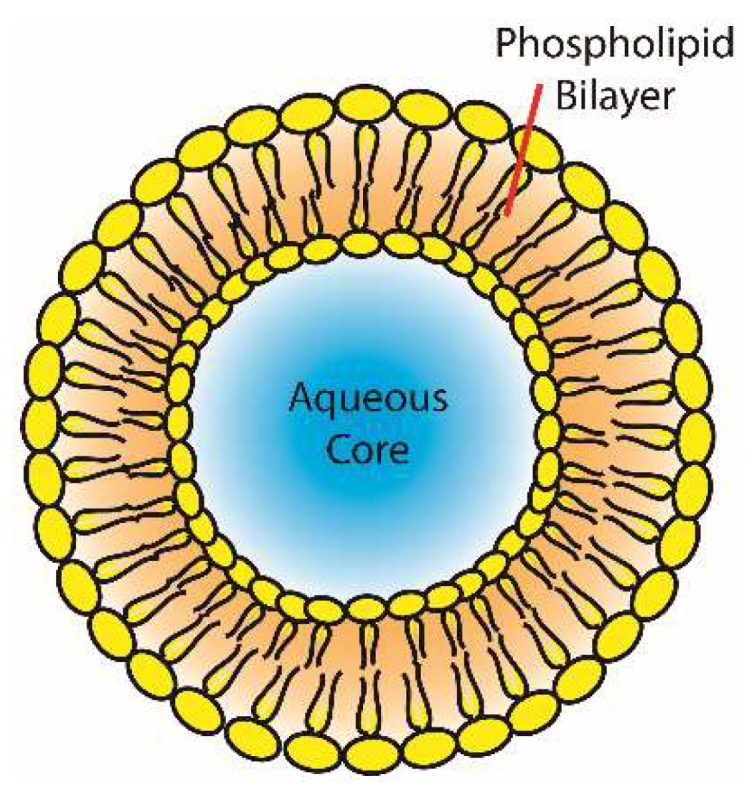
Structure of a liposome. A liposome consists of a phospholipid bilayer with an aqueous core.

**Table 1 pharmaceutics-11-00124-t001:** Strategies developed to optimize liposomal products for pharmaceutical use.

Method Developed	Utility	References
Dehydration-rehydration method	Improve passive encapsulation efficiency	[48]
Modulation of lipid fluidity	Improve passive encapsulation efficiency	[49]
pH gradient loading	Remote loading	[50,51,52,53]
Use of ionophore to load small molecules	Improve remote loading efficiency	[54]
Use of ethanol to load small molecules	Improve remote loading efficiency	[46]
Microencapsulation method	Improve loading efficiency of water soluble and insoluble compound	[55]
Layersomes	Improve liposome stability and oral delivery	[56]
Hyaluronan coating of liposomes	Enable topical applications	[57]
Use of metal ion gradient	Stabilize water-soluble compounds	[58,59]
Metaplex technology	Enable development of poorly soluble metal-binding compounds	[60,61]
Use of cationic lipids	Deliver nucleic acids	[39,40]
Lipid nanoparticle (LNP) technology	Optimize delivery of nucleic acids for clinical use	[62,63,64]

**Table 2 pharmaceutics-11-00124-t002:** Canadian discoveries that were involved in driving the advancement of liposomal pharmaceutical products.

Canadian Discoveries	References
Use of antibodies to mediate targeting with liposomes	[121,122]
Selective targeting of liposomes to the blood compartment	[123]
Use of GM_1_ ganglioside in liposomes, leading to the development of “PEGylation”	[122]
Role of PEG in preventing liposome aggregation	[124]
Development of low-cholesterol liposomes with lipids that prevent aggregation	[124]
Maintenance of the drug-drug ratio for two drugs encapsulated in one liposome	[77]

**Table 3 pharmaceutics-11-00124-t003:** Regulatory approved liposomal formulations.

Approval Year	Trade Name	Active Agent	Lipid Composition	Approved Indication(s)	Current Ownership	References
1993	Epaxal (discontinued)	Inactivated hepatitis A virus (strain RGSB)	DOPC:DOPE (75:25 molar ratio)	Hepatitis A	Janssen Pharmaceuticals	[156,157]
1995	Doxil	Doxorubicin	HSPC:Cholesterol:PEG 2000-DSPE (56:39:5 molar ratio)	Ovarian, breast cancer, Kaposi’s sarcoma	Janssen Pharmaceuticals	[158,159,160]
1995	Abelcet	Amphotericin B	DMPC:DMPG (7:3 molar ratio)	Invasive severe fungal infections	Leadiant Biosciences	[161,162,163]
1996	DaunoXome	Daunorubicin	DSPC:Cholesterol (2:1 molar ratio)	AIDS-related Kaposi’s sarcoma	Galen Pharmaceuticals	[164,165]
1996	Amphotec	Amphotericin B	Cholesteryl sulphate:Amphotericin B (1:1 molar ratio)	Severe fungal infections	Kadmon Pharmaceuticals	[166]
1997	Ambisome	Amphotericin B	HSPC:DSPG:Cholesterol:Amphotericin B (2:0.8:1:0.4 molar ratio)	Presumed fungal infections	Astellas Pharma & Gilead Sciences	[167,168,169]
1997	Inflexal V (recalled)	Inactivated hemaglutinine of Influenza virus strains A and B	DOPC:DOPE (75:25 molar ratio)	Influenza	Crucell, Berna Biotech	[170]
1999	Depocyt (discontinued)	Cytarabine/Ara-C	Cholesterol:Triolein:DOPC:DPPG (11:1:7:1 molar ratio)	Neoplastic meningitis	Pacira Pharmaceuticals	[171,172]
2000	Myocet	Doxorubicin	EPC:Cholesterol (55:45 molar ratio)	Combination therapy with cyclophosphamide in metastatic breast cancer	Teva Pharmaceutical Industries	[173,174]
2000	Visudyne	Verteporfin	EPG:DMPC (3:5 molar ratio)	Choroidal neovascularisation	Cheplapharm Arzneimittel GmbH	[175,176]
2004	DepoDur (discontinued)	Morphine sulfate	Cholesterol:Triolein:DOPC:DPPG (11:1:7:1 molar ratio)	Pain management	Flynn Pharmaceuticals	[177]
2009	Mepact	Mifamurtide	DOPS:POPC (3:7 molar ratio)	High-grade, resectable, non-metastatic osteosarcoma	Takeda Pharmaceutical Ltd.	[178]
2011	Exparel	Bupivacaine	DEPC, DPPG, Cholesterol and Tricaprylin	Pain management	Pacira Pharmaceuticals, Inc.	[179,180]
2012	Marqibo	Vincristine	SM:Cholesterol (55:45 molar ratio)	Acute lymphoblastic leukemia	Spectrum Pharmaceuticals	[181,182]
2015	Onivyde	Irinotecan	DSPC:MPEG-2000:DSPE (3:2:0.015 molar ratio)	Combination therapy with fluorouracil and leucovorin in metastatic adenocarcinoma of the pancreas	Ipsen Biopharmaceuticals	[183]
2017	Vyxeos	Daunorubicin/Cytarabine	DSPC:DSPG:CHOL (7:2:1 molar ratio)	Therapy related acute myeloid leukemia (t-AML) or AML with myelodysplasia-related changes (AML-MRC)	Jazz Pharmaceuticals	[184,185]
2018	Onpattro	Patisiran	Dlin-MC3-DMA, PEG2000-C-DMG	Hereditary transthyretin-mediated amyloidosis	Alnylam Pharmaceuticals, Inc.	[185]

HSPC (hydrogenated soy phosphatidylcholine); PEG (polyethylene glycol); DSPE (distearoyl-sn-glycero-phosphoethanolamine); DSPC (distearoylphosphatidylcholine); DOPC (dioleoylphosphatidylcholine); DPPG (dipalmitoylphosphatidylglycerol); EPC (egg phosphatidylcholine); DOPS (dioleoylphosphatidylserine); POPC (palmitoyloleoylphosphatidylcholine); SM (sphingomyelin); MPEG (methoxy polyethylene glycol); DMPC (dimyristoyl phosphatidylcholine); DMPG (dimyristoyl phosphatidylglycerol); DSPG (distearoylphosphatidylglycerol); DEPC (dierucoylphosphatidylcholine); DOPE (dioleoly-sn-glycero-phophoethanolamine).

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
