# Peer review of "What Drives Innovation: The Canadian Touch on Liposomal Therapeutics"

_pharmaceutics, 2019, doi:10.3390/pharmaceutics11030124_

Round 1

Reviewer 1 Report

The authors present a review that highlights the Canadian contribution to the development of relevant liposomal technologies. They discuss the methods of liposome preparation, to drug loading strategies, storage strategies and targeted delivery, with a special focus on how Canadian investigators and entrepreneurs impacted the field.

More specifically, the authors address the main contributions that stimulated and enriched the innovation in the field (especially in the context of the Canadian contribution), highlighting the main factors that stimulated the development of a successful strategy.

Most of the description are clearly presented and understandable in its present version.

The paper only needs some improvement before being suitable for publication.

The authors might take into account some of the remarks below:

1) it would be better to introduce a numbering of the different paragraphs developed

2) the paragraph (at page 5 - row 188) entitled: “Optimization of Liposomes for Pharmaceutical Use”, describe efficient ways to encapsulate drugs (such as the high pressure extrusion method, dehydration-rehydration method, the pH gradient loading method, and the transition metal-complexation technology (and the Metaplex technology)).

In order to have a summary of the main approaches developed, we suggest the insertion of a Table (indicating the main approaches and the main relevant references).

Moreover, starting from from page 8 - row 334, the authors begin to describe “the extended use of liposomes to nucleic acid delivery”. I suggest in this case to insert this description within a sub-paragraph.

3) the paragraph (at page 5 – row 361) entitled: Other key Canadian discoveries that impacted the development of therapeutically interesting drugs” discuss the …conjugation of probes that could mediate selective cellular uptake to liposome surface (such as (monoclonal )antibodies, peptide-mediated targeting, polyethylene glycol (PEG).

Also in this case, I suggest the insertion of a Table (indicating the main approaches and the main relevant references).

Moreocer, starting from page 10 - row 426, the authors begin to describe “the strategy to encapsulate multiple drugs in the same liposome”. Also in this case, I suggest to insert this description within a sub-paragraph.

Author Response

Response to Reviewer 1 Comments

Point 1: It would be better to introduce a numbering of the different paragraphs developed

Response 1: We thank the reviewer for this suggestion to better organize the manuscript. We have now added a numbering system for the various topics and sub-topics.

Point 2: the paragraph (at page 5 - row 188) entitled: “Optimization of Liposomes for Pharmaceutical Use”, describe efficient ways to encapsulate drugs (such as the high pressure extrusion method, dehydration-rehydration method, the pH gradient loading method, and the transition metal-complexation technology (and the Metaplex technology)).

In order to have a summary of the main approaches developed, we suggest the insertion of a Table (indicating the main approaches and the main relevant references).

Moreover, starting from from page 8 - row 334, the authors begin to describe “the extended use of liposomes to nucleic acid delivery”. I suggest in this case to insert this description within a sub-paragraph.

Response 2: We thank the reviewer for the suggestion and have made the correction accordingly. A table listing the main approaches developed to optimize liposomes for pharmaceutical development has been created (please see Table 1 on pg. 5).

Point 3: the paragraph (at page 5 – row 361) entitled: Other key Canadian discoveries that impacted the development of therapeutically interesting drugs” discuss the …conjugation of probes that could mediate selective cellular uptake to liposome surface (such as (monoclonal )antibodies, peptide-mediated targeting, polyethylene glycol (PEG).

Also in this case, I suggest the insertion of a Table (indicating the main approaches and the main relevant references).

Moreover, starting from page 10 - row 426, the authors begin to describe “the strategy to encapsulate multiple drugs in the same liposome”. Also in this case, I suggest to insert this description within a sub-paragraph.

Response 3: We thank the reviewer for these suggestions, again, to better organize the manuscript. We have now made these changes and have inserted a table summarizing some of the key Canadian inventions that impacted pharmaceutical development of liposomal products (please see Table 2; pg. 9).

Reviewer 2 Report

The contribution submitted by Bally and co-workers is an inspirational success-story on liposomal technology covering decades of prolific and visionary research in Canada. It provides the reader with an extensive and careful overview of the debut of liposomes as drug delivery carriers till the most pioneering applications, passing through the many commercial successes in the pharmaceutical market. A valuable review, who can certainly serve as example and source of precious references for the youngest generation of liposomal scientists.

Minor comments:

Table 1: it would be interesting to add an extra column in the table reporting some original milestone academic references for each of the approved liposomal formulations listed

Typos:

page 6 lines 257-258: irionotecan, irionphore

Page 7 lines 284: anti-alchololic

Author Response

Response to Reviewer 2 Comments

Point 1: Table 1: it would be interesting to add an extra column in the table reporting some original milestone academic references for each of the approved liposomal formulations listed

Response 1: We thank the reviewer for the suggestion. We have added a column to the table highlighting some of the academic and clinical milestones achieved which led to the approval of the listed liposomal formulations.

Point 2: Typos:

page 6 lines 257-258: irionotecan, irionphore

Page 7 lines 284: anti-alchololic

Response 2: We thank the reviewer for identifying these errors. We have corrected them accordingly.

Reviewer 3 Report

The document is well worded and remains on topic. There are a couple points perhaps:

It is unclear to me why figure 2 is listed before figure 1 though.

It would also be nice in figure 2, so perhaps have a label on each arrow/timeline for clarity.

Use of the Oxford comma would be advisable throughout.

Spacing before references is not always consistent.

For consistency; please either apply Dr. to all or none (where applicable of course).

In line 191 it is stated '...,given an optimal size...'; what determined this optimal size and what is it optimal for? Also, spacing is off in the same sentence for L/mole (may just be formatting in my document though)

Line 260; in regard to Onivyde. Although Canada was 'too slow', it can be argued that preclinical research for the MM-398 was carried out in Canada.

Line 318 is unsubstantiated. Please insert a reference.

Line 336: typo. 'dsigned'.

Lines 369-372; which imaging modalities were used and which isotopes? (while it is true that all the info is in the references, it would still be nice to have this information in this document).

Line 396 would be better places when mentioning Matsumurra and Maeda in line 375.

Line 435 : in vitro should be italicized (please check entire document).

Author Response

Response to Reviewer 3 Comments

Point 1: It is unclear to me why figure 2 is listed before figure 1 though.

Response 1: We agree with the reviewer that Figure 1 should come before Figure 2. We believe this was a result of the formatting changes performed by the Editorial Office. As such, we have requested for the figures to be placed in accordance to the in-text reference.

Point 2: It would also be nice in figure 2, so perhaps have a label on each arrow/timeline for clarity.

Response 2: We agree with the reviewer and have added a label to each arrow on the timeline.

Point 3: Use of the Oxford comma would be advisable throughout.

Response 3: We have revised the manuscript as suggested by the reviewer.

Point 4: Spacing before references is not always consistent.

Response 4: We thank the reviewer for pointing out the inconsistency and have now checked the spacing for all references.

Point 5: For consistency; please either apply Dr. to all or none (where applicable of course).

Response 5: We have now revised the manuscripts such that last names only are mentioned when describing published articles and “Dr.” is applied where specific scientists or investigators are mentioned. We hope this would make the manuscript more consistent.

Point 6: In line 191 it is stated '...,given an optimal size...'; what determined this optimal size and what is it optimal for? Also, spacing is off in the same sentence for L/mole (may just be formatting in my document though)

Response 6: We apologize for the lack of clarity. We have now revised the manuscript to explain that the optimal liposome size of 100-200 nm refers to nanomedicines that attempt to leverage the enhanced permeability and retention effect (Lines 197-200). We have also adjusted the spacing issue associated with the rest of that particular sentence.

Point 7: Line 260; in regard to Onivyde. Although Canada was 'too slow', it can be argued that preclinical research for the MM-398 was carried out in Canada.

Response 7: We respectfully disagree with the reviewer that MM-398 was studied pre-clinically in Canada as published literature indicate that the formulation was extensively tested by Merrimack Pharmaceuticals in the United States.

Point 8: Line 318 is unsubstantiated. Please insert a reference.

Response 8: We apologize for not including a reference in line 318. We have now included references that would describe Cuprous’ technology (Line 326).

Point 9: Line 336: typo. 'dsigned'.

Response 9: The typo has been corrected.

Point 10: Lines 369-372; which imaging modalities were used and which isotopes? (while it is true that all the info is in the references, it would still be nice to have this information in this document).

Response 10: We have now revised the manuscript to better describe the imaging modalities and isotopes used (Lines 379-388). We thank the reviewer for this suggestion.

Point 11: Line 396 would be better places when mentioning Matsumurra and Maeda in line 375.

Response 11: We agree with the reviewer and have made the correction accordingly.

Point 12: Line 435 : in vitro should be italicized (please check entire document).

Response 12: We have corrected the formatting of “in vitro” throughout the entire document as suggested by the reviewer.

Reviewer 4 Report

The paper “What drives innovation: the canadian touch on liposomal therapeutics” reviews the Canadian contribution to the development of liposomal technologies. The review is well written organized. My only suggestion is to add some more references in “Brief History” along with ref. n. 22 on liposomal lidocaine: Penetration Enhancer-containing Vesicles: does the penetration enhancer structure affect topical drug delivery? Current Drug Targets, 2015, 16, 1438-1447; Exploring the co-loading of lidocaine chemical forms in surfactant/phospholipid vesicles for improved skin delivery, Journal of Pharmacy and Pharmacology, 67, pp. 909–917.

Author Response

Response to Reviewer 4 Comments

Point 1: The paper “What drives innovation: the canadian touch on liposomal therapeutics” reviews the Canadian contribution to the development of liposomal technologies. The review is well written organized. My only suggestion is to add some more references in “Brief History” along with ref. n. 22 on liposomal lidocaine: Penetration Enhancer-containing Vesicles: does the penetration enhancer structure affect topical drug delivery? Current Drug Targets, 2015, 16, 1438-1447; Exploring the co-loading of lidocaine chemical forms in surfactant/phospholipid vesicles for improved skin delivery, Journal of Pharmacy and Pharmacology, 67, pp. 909–917.

Response 1: We thank the reviewer for the suggestion. We have revised the manuscript accordingly.